APCTP Pre2025 - 011

# Towards holographic color superconductivity in QCD

Jesús Cruz Rojas[1*], Tuna Demircik[2†], Christian Ecker[3‡], Matti Järvinen[4,5♯]

**1** Departamento de Física de Altas Energías, Instituto de Ciencias Nucleares, Universidad Nacional Autónoma de México, Apartado Postal 70-543, CDMX 04510, México

**2** Institute for Theoretical Physics, Utrecht University, Leuvenlaan 4, 3584 CE Utrecht, The Netherlands

**3** Institute for Theoretical Physics, Goethe University 60438, Frankfurt am Main, Germany

**4** Asia Pacific Center for Theoretical Physics, Pohang, 37673, Korea

**5** Department of Physics, Pohang University of Science and Technology, Pohang, 37673, Korea

\* jesus.cruz@correo.nucleares.unam.mx

† t.demircik@uu.nl

‡ ecker@itp.uni-frankfurt.de

♯ matti.jarvinen@apctp.org

May 27, 2025

## Abstract

We extend the holographic V-QCD model by introducing a charged scalar field sector to represent the condensation of paired quark matter in the deconfined phase. By incorporating this new sector into the previously established framework for nuclear and quark matter, we obtain a phase diagram that, in addition to the first-order deconfinement transition and its critical end-point, also features a second-order transition between paired and unpaired quark matter. The critical temperature for quark pairing exhibits only a mild dependence on the chemical potential and can reach values as high as $T_{\mathrm{crit}} \approx 30$ MeV. Comparison of the growth rate for the formation of homogeneous paired phases to the growth rate of previously discovered modulated phases suggests that the former is subdominant to the latter.

# 1   Introduction

Quantum Chromodynamics (QCD), the fundamental theory governing the strong interaction, exhibits a complex phase structure that remains only partially understood. Despite the analysis being challenging in the regime of finite baryon density and strong coupling, some general features are generally accepted. At low densities and temperatures, QCD is strongly coupled leading to confinement of its fundamental degrees of freedom, quarks and gluons, within hadrons. First-principles lattice QCD simulations at zero chemical potential and finite temperatures predict a smooth crossover between the confined and deconfined phases at a pseudo-critical temperature of approximately 158 MeV [1, 2], a finding supported by relativistic heavy-ion collision experiments (see [3] for recent review).

Substantial experimental and theoretical efforts are currently underway to find the critical endpoint of a suspected first order deconfinement transition at finite baryon chemical potential in the phase diagram [4–8]. Moreover, due to asymptotic freedom, QCD becomes weakly coupled at large momentum transfers, implying that in regimes of high densities a perturbative description of deconfined quarks and gluons is applicable [9]. Additionally, it is well established [10, 11] that an attractive interaction between fermions leads to their condensation. Since the strong force includes such attractive interactions among color-charged particles, quarks are predicted to condense at high densities. Consequently, at sufficiently low temperatures and high densities, deconfined quark matter is expected to settle into its ground state as a color superconducting condensate, in which quarks pair up [12–18]. In addition, these color superconducting phases may compete with other "exotic" phases such as quarkyonic phases [19–21], and spatially modulated phases [22–26].

The only plausible environments for the existence of such exotic phases are extreme astrophysical settings, such as the cores of neutron stars or the remnants of their binary mergers. While this possibility remains speculative, recent, and anticipated future, advances in multi-messenger astrophysics, particularly through observations of gravitational waves and their electromagnetic counterparts, provide promising avenues to address the

question [27, 28]. In light of these developments, robust predictions for the behavior of condensed quark matter are urgently needed. For example, it is crucial to determine whether the conditions inside neutron stars and their mergers can, in principle, support the existence of such matter. In particular, it is important to determine whether the transition temperature from unpaired to paired quark matter is sufficiently high to allow its formation in astrophysical environments with temperatures of several tens of MeV, or so high that such formation becomes unavoidable under these conditions (see, e.g., [29–31] for reviews). Furthermore, the presence of condensed quark matter alters the most fundamental ingredient for the thermodynamic description of the system, the equation of state (EOS). The EOS, combined with Einstein's field equations, governs the mass–radius relation of stars, the merger and post-merger dynamics of binary systems, and whether they can withstand gravitational collapse into a black hole [32–37].

First-principles methods, such as lattice QCD, are not well-suited for addressing this question due to the fermionic sign problem. Recent analyses in perturbative QCD have provided constraints on the EOS at densities beyond those realized in neutron star interiors [38, 39] and have derived bounds on the superconducting gap from neutron star observations [40]. Promising alternatives include non-perturbative continuum approaches like Dyson–Schwinger Equations [41–46] and the Functional Renormalization Group [25,38,47], although further development is necessary before they can be directly applied to neutron star phenomenology. Currently, the state-of-the-art approach to investigating color superconductivity under neutron star conditions is the QCD-inspired Nambu-Jona-Lasinio (NJL) model (see [48] for a review). This model enables the systematic identification of various paired phases within the phase diagram and facilitates the study of their presence in neutron stars (see, e.g., [49] and references therein).

While the aforementioned approaches are either directly related or close to QCD, their application to neutron star matter comes with additional difficulties. For example, NJL is an effective model for quark matter only, neglecting gluon interaction and confinement. Including gluon dynamics therefore requires extending the model [50,51], and for neutron star applications, NJL models need to be combined with separate descriptions for nuclear matter. Furthermore the model is known to be non-renormalizable and therefore suffers from cutoff artifacts. Although the problem has been partially mitigated in [52] by imposing renormalization group consistency [53], convergence of the model to perturbative QCD at asymptotic density is neither expected nor guaranteed unlike in model independent approaches where this can be imposed as a constraint (see, e.g., [54] and references therein).

Motivated by this, we explore an alternative approach called holographic QCD, which is a non-perturbative method rooted in string theory. It allows us to obtain thermodynamic quantities of strongly coupled gauge theories that are similar to QCD by solving their higher-dimensional gravity dual. Since the exact holographic dual of QCD remains unknown, several simplifying assumptions must be adopted. Standard examples where a precise correspondence has been established typically include superconformal field theories in the large $N_c$ limit, which admit a dual description in terms of classical gravity. However, QCD is neither supersymmetric nor conformal, but possesses a discrete spectrum and exhibits confinement. Moreover QCD has $N_c = 3$ which is far from being considered large, and the coupling constant flows becoming small at high energies (asymptotic freedom). Taking these caveats into account, the holographic approach has provided valuable insights into the strongly coupled dynamics of quark-gluon plasma in relativistic heavy-ion collisions [55], strongly correlated condensed matter systems [56, 57], and quantum states on dynamical backgrounds relevant to cosmology [58, 59]. Most importantly, and crucial for the context of this work, this method bypasses several limitations inherent in

the aforementioned non-holographic approaches. For example, it allows one to integrate both baryon and quark matter sectors within a single model, providing direct access to the deconfinement phase transition, it also allows for a simple mechanism of chiral symmetry breaking (see reviews [60,61] and references therein for recent applications to neutron star matter).

However, the holographic description of the condensed quark matter phase remains far from fully understood. Several different approaches have been pursued. Early attempts were mostly based on top-down constructions, such as in [62–65], focusing on known string theory duals of supersymmetric cousins of QCD. The advantage of this approach is that these string theory duals possess a single-trace, gauge-invariant order parameter, allowing the study of color superconducting phases through the spontaneous breaking of the local color symmetry group. Moreover, an important limitation of these models is that they involve only small, partial color Higgsing – that is, only a subset of the original color gauge symmetry is broken by the formation of diquark condensates, giving mass to just some of the gluons while leaving the rest of the gauge sector unbroken and massless.

Another approach uses bottom-up constructions [66–73], which adopt the assumption that the color group can be treated as a global symmetry. In this approach, a quark-quark condensate phase is modeled by considering an additional sector in the action with a vectorially charged scalar field that mimics a color superconducting phase of QCD at finite densities and temperatures. While this setup does not fully capture all the features of the color superconducting state which requires spontaneous breaking of local $SU(N_c)$, it effectively encodes the essential physics of spontaneous symmetry breaking and condensate formation, providing an analogy to superfluid-like superconductivity.

A recent study [74] proposes a bottom-up model where color symmetry is treated as a local gauge symmetry, aiming to circumvent some technical difficulties of the top-down approach, such as the limitation to small partial Higgsing. Nevertheless, all these early studies suffer from a serious deficiency: none have been precisely calibrated against QCD phenomenology. In this work, we aim to take the next step toward bridging this gap.

We work with V-QCD, a class of holographic models [75], which are particularly suitable for the description of matter at nonzero temperature and density [76–78] (for a top-down alternative, see recent work on the Witten-Sakai-Sugimoto model [79–82]). These models can be accurately fitted to lattice data [83] at small density, can be extended to include both nuclear [78,84,85] and quark matter [77,83,86] at higher density. Interestingly, this setup is also in agreement with known constraints (e.g., from the constraints coming from neutron star measurements) in this high density region [87–89]. Consequently, it was possible to construct a state-of-the-art model for the QCD equation of state, covering the whole phase diagram apart from asymptotically high densities and pressures, by using V-QCD in combination with nuclear theory models [90]. In particular, this model covers the region relevant for binary neutron star mergers [37, 91–93], but also extend to the low-density region relevant for heavy-ion collisions.

As a first step toward a phenomenologically viable description of color superconducting phases within the V-QCD framework, in this article, we study quark condensation by adding a charged scalar sector. Our construction follows the "holographic superconductor" [94, 95] model, which also means that it is a global $U(1)$ that is broken by the condensate on the field theory side, while breaking of the gauge symmetry is not described[1]. This also means that, similarly to the earlier bottom-up approaches (e.g. [68, 70]) we do not attempt to describe the condensation and symmetry breaking structure of any specific color superconductor phase. However, we note that the $U(1)$, which is broken, is actually

---

[1]Because of this, the model is closer to a holographic superfluid rather than a superconductor (see, e.g., [96]).

the baryon number. Therefore our approach can be interpreted to be close to the color-flavor-locked phase [14], which similarly breaks the baryon number (rather than locking it with gauge symmetry) and is a superfluid. Breaking of the baryon number is also a desired feature in the holographic model, because in this case the condensed hairy black holes will have zero entropy [97]. This removes the phenomenological issue that the V-QCD model (with unpaired quark matter) has nonzero entropy in the zero temperature limit [77].

We start by analysing the instabilities of the unpaired quark matter in this article. That is, we focus on the chirally symmetric, deconfined phase at finite temperature and finite baryon density. At low temperatures the geometry becomes approximately $\text{AdS}_2 \times \mathbb{R}^3$, which is known to enhance instabilities [77, 98–100]. Tuning the parameters in the corresponding action one can identify an instability towards the condensation of the charged scalar field caused by the change in its effective five-dimensional mass, in part caused by the coupling to the charge of the black hole background. We then compare the quark condensation instability with the spatially modulated instabilities found in [101, 102], which arise due to a Chern-Simons (CS) term on the higher dimensional gravity side. Finally, we also study the phase where the charged scalar has condensed by using a probe approximation.

The rest of the article is organized as follows. In Section 2, we describe the holographic model used to study the deconfined matter phase. We first describe the unpaired sector and explain how the potentials in the action are fixed to reproduce salient features of QCD. We then describe the paired quark matter sector. In Section 3, we begin by discussing the Breitenlohner-Freedman (BF) bound and the general mechanism that drives the system toward instability. We subsequently focus on the pairing instability and the spatially modulated instability in the following subsections. In Section 4, we explain how we calculate the free energy of the condensed phase as a function of temperature and chemical potential. Section 5 presents our results, including the phase diagram, the EOS, and the stability analysis. Finally, we summarize and conclude in Section 6. Additional details regarding the instabilities at the zero-temperature $\text{AdS}_2$ point, both for the pairing and the modulated sectors, are provided in Appendix A.

## 2 Setup

### 2.1 The V-QCD framework

The V-QCD setup [75] is a class of holographic models for QCD, inspired by five dimensional noncritical string theory but refined and generalized using a bottom-up approach to achieve the desired phenomenology. It consists of two main building blocks: the glue sector and the flavor sector, with full backreaction included. The gluon sector is described by improved holographic QCD [103, 104], while the flavor sector is modeled by incorporating the tachyonic Dirac-Born-Infeld (TDBI) and CS actions [105, 106]. The string theory motivation of the model [75] considers the Veneziano limit [107], i.e., the limit where $N_c$ and $N_f$ are taken to infinity while keeping the ratio $x = N_f/N_c$ fixed. In this limit, the flavor sector is fully backreacted to the gluon sector. Inspired by QCD, we set $x = 1$ in what follows.

The version of the V-QCD action used here is given by a sum of four terms:

$$S = S_{\text{g}} + S_{\text{TDBI}} + S_{\text{CS}} + S_{\psi}. \tag{1}$$

We will briefly outline the fundamental components of the actions for the glue sector ($S_{\text{g}}$), the flavor sector ($S_{\text{TDBI}}$) and the Chern-Simons sector ($S_{\text{CS}}$) below, while deferring the

detailed discussion of the action describing the condensed quark matter phase ($S_\psi$) to Sec. 2.4.

The action for the gluon sector is described by five-dimensional dilaton-gravity which provides the dual description for $SU(N_c)$ gauge theory:

$$S_{\text{g}} = M_p^3 N_c^2 \int d^5x \sqrt{-g} \left[ R - \frac{4}{3} g^{MN} \partial_M \phi \partial_N \phi + V_g(\phi) \right] , \tag{2}$$

where $M_p$ is the Planck mass. The five-dimensional metric on the gravity side $g_{MN}$ is dual to the expectation value of the energy-momentum tensor $\langle T^{\mu\nu} \rangle$ of the dual filed theory. We use notation where five (four) dimensional Lorentz indices are denoted by capital Latin (Greek) letters. The dilaton field, $\phi$ is dual to the expectation value of the $G^{\mu\nu} G_{\mu\nu}$ operator, where $G^{\mu\nu}$ is the gluon field in the dual theory. Note that the corresponding source in the field theory is the gauge coupling $g$. In practice this implies that near the boundary, the exponential of the dilaton $\lambda = e^\phi$ is identified as the 't Hooft coupling $g^2 N_c$ (see [103, 104] for details).

For the flavor sector, the global chiral symmetry $U(N_f)_L \times U(N_f)_R$ of QCD is promoted to a gauge symmetry on the gravity side. Since the focus of this article is on the chirally symmetric deconfined phase, the tachyon field ($T^{ij}$) dual to the quark bilinear ($\bar{\psi}^i \psi^j$) vanishes. Consequently, the TDBI action reduces to

$$\begin{aligned} S_{\text{TDBI}} = -\frac{1}{2} M_p^3 N_c \int d^5x \, V_f(\phi) \times \\ \times \text{Tr} \left( \sqrt{-\det(g_{MN} + w(\phi) F_{MN}^{(L)})} + \sqrt{-\det(g_{MN} + w(\phi) F_{MN}^{(R)})} \right) , \end{aligned} \tag{3}$$

where $F_{MN}^{(L/R)}$ is the field strength tensor for the left-/right-handed gauge field $A_M^{ij(L/R)}$, which are $N_f \times N_f$ matrices in flavor space. The fields $A_M^{ij(L/R)}$ are dual to the left-/right-handed flavor currents, $\bar{\psi}^i (1 \pm \gamma_5) \gamma_\mu \psi^j$ of QCD. The determinant is over Lorentz indices and the trace Tr is over flavor indices $i, j = 1 \ldots N_f$.

The CS sector is governed by the flavor anomalies of QCD, but it does not influence the construction of the background solution for the unpaired quark matter phase. However, as shown in [101, 102], fluctuations of the gauge fields are affected by the CS action and become unstable at finite momenta, signaling the emergence of a spatially modulated phase. Remarkably, this instability is largely model-independent and spans a significant region in the phase diagram [102]. This observation implies that the condensed quark matter phase investigated in this work competes with spatially modulated phase driven by the CS action. Accordingly, we explicitly include the CS action

$$\begin{aligned} S_{\text{CS}} = \frac{iN_c}{24\pi^2} \int \text{Tr} \bigg[ &-i A_L \wedge F_L \wedge F_L + \frac{1}{2} A_L \wedge A_L \wedge A_L \wedge F_L + \\ &+ \frac{i}{10} A_L \wedge A_L \wedge A_L \wedge A_L \wedge A_L + i A_R \wedge F_R \wedge F_R - \frac{1}{2} A_R \wedge A_R \wedge A_R \wedge F_R - \\ &- \frac{i}{10} A_R \wedge A_R \wedge A_R \wedge A_R \wedge A_R \bigg] , \end{aligned} \tag{4}$$

with the field strength defined as $F_{L/R} = dA_{L/R} - iA_{L/R} \wedge A_{L/R}$. In the chirally symmetric phase considered here, i.e., when the tachyon vanishes, the form of $S_{\text{CS}}$ in equation (4) is the standard one [106, 108], while the tachyon dependence in more general cases has been recently analyzed in [109].

## 2.2 Unpaired quark matter solution

We focus on the chirally symmetric deconfined phase at finite temperature and baryon number density. In this subsection, we briefly discuss the relevant type of V-QCD background (see [76, 77]).

We first switch to the basis with vectorial and axial gauge fields

$$V_M = \frac{1}{2}\left(A_M^{(L)} + A_M^{(R)}\right), \qquad A_M = \frac{1}{2}\left(A_M^{(L)} - A_M^{(R)}\right). \tag{5}$$

For the finite density background solution, it is enough to consider the flavor singlet vectorial field,

$$V_M^{ij} = \hat{V}_M \delta^{ij}\ , \tag{6}$$

whereas the axial components are set to zero. Restricting to the flavor singlet vectorial terms, the flavor Lagrangian (3) becomes

$$S_{\text{TDBI}}^{\text{bg}} = -xM_p^3 N_c^2 \int d^5x\, V_f(\phi)\sqrt{-\det\left[g_{MN} + w(\phi)\hat{F}_{MN}\right]}, \qquad x = N_f/N_c, \tag{7}$$

where the $\hat{F}_{MN}$ is the field strength tensor of the vectorial singlet field and the ratio $x = N_f/N_c$ is the measure of the backreaction of the flavor sector to glue sector. Note that the CS term (4) vanishes for this background.

The Ansatz for the background metric is a five-dimensional (asymptotically AdS) charged black hole solution

$$ds^2 = e^{2A(r)}\left[f(r)^{-1}dr^2 - f(r)dt^2 + d\mathbf{x}^2\right]\ , \tag{8}$$

where $A(r)$ and $f(r)$ are the warp and the blackening factors respectively. The coordinate $r$ runs from the boundary at $r = r_b$ (which can be set to zero thanks to invariance under shifts in $r$) to a horizon at $r = r_h$. At the boundary, we have that $f(0) = 1$ while the blackening factor vanishes at the horizon $f(r_h) = 0$. In the Ansatz for the background, the only nonvanishing component is the temporal component of the vectorial gauge field which gives rise to a quark number chemical potential on the boundary:

$$\hat{V}_t(r, x^\mu) = \Phi(r), \qquad \mu_q = \Phi(r = 0)\ . \tag{9}$$

We also define the baryon number chemical potential as $\mu_b = N_c\mu_q$, where we will eventually set $N_c = 3$.

Then the background is described by the metric functions $A$ and $f$, the dilaton field $\phi$, and the temporal component of the gauge field $\Phi$ which depend solely on the holographic coordinate $r$. With these assumptions, the action for the background includes only the $r$-derivative of $\Phi$, implying that the equation of motion of $\Phi$ integrates to

$$\frac{1}{M_p^3 N_c N_f}n_q = -\frac{1}{M_p^3 N_c N_f}\frac{\partial \mathcal{L}_{TDBI}}{\partial \Phi'} = -\frac{e^A V_f(\phi)w(\phi)^2\Phi'}{\sqrt{1 - e^{-4A}w(\phi)^2(\Phi')^2}}\ , \tag{10}$$

where $\mathcal{L}_{\text{TDBI}}$ is the Lagrangian of the flavor action (7) and the constant $n_q$ is identified as the quark number density. This equation can be solved for $\Phi'$ and integrated to obtain the chemical potential $\mu_q$ [77].

The features of the background in the weak coupling region, i.e., the ultraviolet (UV) limit, are determined entirely by the combination of $V_g$ and $V_f$. Expanding this combination in a Taylor series around $\lambda = e^\phi = 0$, one obtains

$$V_g(\phi) - xV_f(\phi) = \frac{12}{\ell^2}\left[1 + v_1\frac{e^\phi}{8\pi^2} + v_2\left(\frac{e^\phi}{8\pi^2}\right)^2 + \mathcal{O}\left(e^{3\phi}\right)\right]\ . \tag{11}$$

The constant term of $12/\ell^2$ ensures that the geometry is asymptotically AdS$_5$, with $\ell$ being the AdS radius. As a consequence of this expansion, the near-boundary behavior of the geometry receives logarithmic corrections [103]:

$$A(r) = -\log\frac{r}{\ell_0} + \frac{4}{9\log(r\Lambda)} + \tag{12}$$

$$+ \frac{\left(\frac{95}{162} - \frac{32v_2}{81v_1^2}\right) + \left(-\frac{23}{81} + \frac{64v_2}{81v_1^2}\right)\log(-\log(r\Lambda))}{(\log(r\Lambda))^2} + \mathcal{O}\left(\frac{1}{(\log(r\Lambda))^3}\right) \,,$$

$$\frac{v_1 e^{\phi(r)}}{8\pi^2} = -\frac{8}{9\log(r\Lambda)} + \frac{\left(\frac{46}{81} - \frac{128v_2}{81v_1^2}\right)\log(-\log(r\Lambda))}{(\log(r\Lambda))^2} + \mathcal{O}\left(\frac{1}{(\log(r\Lambda))^3}\right) \,. \tag{13}$$

As expected, the source term of the dilaton flows logarithmically instead of being a mere constant. The value of the source is now identified with the scale $\Lambda = \Lambda_{\text{UV}}$ and all dimensionful quantities are henceforth expressed in units of $\Lambda_{\text{UV}}$. Interpreting the warp factor $A(r)$ as logarithm of the energy scale allows for direct mapping of the coefficients $v_i$ to the $\beta$-function of QCD [75, 103, 104].

At strong coupling, the infrared (IR) geometry of the background solutions for the unpaired quark matter phase at finite density ($\mu_q \sim \Lambda$) depends on the temperature. Specifically, the IR geometry is characterized as follows [77, 110]:

- $T = 0$: The geometry asymptotically approaches AdS$_2 \times \mathbb{R}^3$.

- $0 < T \ll \Lambda$: The pure AdS$_2$ geometry is replaced by an AdS$_2$ black hole.

- $T \sim \Lambda$: The geometry corresponds to a "regular" charged black hole.

The emergence of the AdS$_2$ geometry in the zero temperature limit signals the presence of a "quantum critical" region [98–100], and is essential for the instabilities towards condensed phases as we shall see below.

## 2.3 Choice of the V-QCD potentials

The action for the background (2) and (7) includes dilaton-dependent potentials: $V_g(\phi)$, $V_f(\phi)$ and $w(\phi)$. These potentials are chosen to reproduce the salient features of QCD in both the weak coupling and strong coupling regimes. Since the exponential of the dilaton is identified as the 't Hooft coupling near the boundary, it encodes the running of the coupling on the QCD side. Consequently, this allows us to fix the asymptotic behavior of the potentials separately at small coupling ($\phi \to -\infty$) [75, 103] and at large coupling ($\phi \to \infty$) [75, 78, 104, 111, 112], as we now explain.

In the UV, the asymptotics are determined by matching the holographic RG flow to perturbative QCD predictions for the $\beta$-function (as outlined in Section 2.2 which discusses the features of the background geometry) [103], as well as to the running of the quark mass [75]. In the IR, the asymptotics are fixed by requiring consistency with key features of QCD, namely confinement, chiral symmetry breaking, gapped glueball and meson spectra, linear trajectories in squared masses of the radial excitations, and a qualitatively reasonable phase diagram [78, 104, 113, 114]. Interestingly, the IR asymptotics following from these requirements is consistent with the potentials being flat (up to polynomial corrections in $\phi$) in the string frame [78, 113]. That is, substituting $g = e^{-4\phi/3}g_{\text{s}}$ in the V-QCD action, as required for the transformation between the Einstein and string frames in five dimensions, the exponential factors cancel at large $\phi$, except for the usual

$e^{-2\phi}$ and $e^{-\phi}$ factors in the gravity and DBI actions, respectively. This means that the Einstein frame potentials behave as[2]

$$V_g(\phi) \sim e^{4\phi/3} \ , \qquad V_f(\phi) \sim e^{7\phi/3} \ , \qquad w(\phi) \sim e^{-4\phi/3} \ , \qquad (14)$$

as $\phi \to +\infty$.

These UV and IR asymptotics are then interpolated by fitting with lattice data for large-$N_c$ pure Yang-Mills [115] and for $N_c = 3$ QCD with $N_f = 2+1$ flavors [116] at small $n_q$. Notably, once the UV and IR asymptotics are fixed to match QCD features, the fit to lattice data becomes highly constrained, but a good fit to lattice data is possible despite this [83, 86, 117–121]. However, even after the fit to lattice data there is still some freedom left in choosing the potentials. That is, one is left with essentially a one-parameter family of choices [78, 83]. In this work, we use the 7a potential set (an intermediate choice) [83]. The functional form of the potentials is

$$V_g(\phi) = 12 \left[ 1 + V_1 e^{\phi} + \frac{V_2 e^{2\phi}}{1 + e^{\phi}/\lambda_0} + V_{\mathrm{IR}} \frac{(e^{\phi}/\lambda_0)^{4/3}}{e^{\lambda_0/e^{\phi}}} \sqrt{\log(1 + e^{\phi}/\lambda_0)} \right] , \qquad (15)$$

$$V_f(\phi) = W_0 + W_1 e^{\phi} + \frac{W_2 e^{2\phi}}{1 + e^{\phi}/\lambda_0} + W_{\mathrm{IR}} \frac{(e^{\phi}/\lambda_0)^2}{e^{\lambda_0/e^{\phi}}} , \qquad (16)$$

$$\frac{1}{w(\phi)} = w_0 \left[ 1 + \bar{w}_0 e^{-\hat{\lambda}_0/e^{\phi}} \frac{(e^{\phi}/\hat{\lambda}_0)^{4/3}}{\log(1 + e^{\phi}/\hat{\lambda}_0)} \right] , \qquad (17)$$

where most UV parameters are determined by comparison with the perturbative RG flow:

$$V_1 = \frac{11}{27\pi^2} \ , \qquad V_2 = \frac{4619}{46656\pi^4} \ , \qquad W_1 = \frac{8 + 3W_0}{9\pi^2} \ , \qquad W_2 = \frac{6488 + 999W_0}{15552\pi^4} \ . \qquad (18)$$

The remaining UV parameter $W_0$ reflects the freedom left after fitting to lattice data, and it is set to $W_0 = 2.5$ for potentials 7a, while the the remaining parameters are determined by comparison with lattice results

$$\lambda_0 = 8\pi^2/3 \ , \qquad\qquad 8\pi^2/\hat{\lambda}_0 = 1.18 \ , \qquad\qquad V_{\mathrm{IR}} = 2.05 \ , \qquad (19)$$

$$W_{\mathrm{IR}} = 0.9 \ , \qquad\qquad w_0 = 1.28 \ , \qquad\qquad \bar{w}_0 = 18 \ , \qquad (20)$$

$$\Lambda_{\mathrm{UV}}/\mathrm{MeV} = 211 \ , \qquad\qquad 180\pi^2 M_p^3 \ell^3/11 = 1.32 \ . \qquad (21)$$

These choices also set the AdS radius to $\ell = 1/\sqrt{1 - W_0/12}$.

## 2.4    Paired quark matter sector

To model the condensed quark matter phase, we follow the prescription for constructing a holographic superconductor as given in [94, 95, 122]. In this approach, a charged bulk scalar is introduced. Through an IR instability mechanism (which will be elaborated in Section 3), the scalar field condenses in the bulk, breaking the bulk $U(1)$ gauge symmetry. This corresponds to the spontaneous breaking of a global $U(1)$ symmetry associated with baryon (quark) number conservation in the boundary field theory [122]. Although this setup does not provide a complete description of the expected condensed quark matter phase—namely, the color superconducting state—it is sufficient to capture the key physics

---

[2]To be precise, agreement with QCD features requires that $V_f \sim e^{v_f\phi}$ with $4/3 < v_f < 10/3$, which includes both the value $v_f = 7/3$ obtained by transforming to the string frame and the value $v_f = 2$ we use below.

of spontaneous symmetry breaking and condensate formation, offering a description analogous to superfluid-like superconductivity. In this sense it is close to the color-flavor-locked phase which also completely breaks the baryon number symmetry and is a superfluid. However, we stress that we do not attempt to construct a precise dual of any specific color superconductor phase in this article, and the charged scalar field should be interpreted to be a dual to some unspecified bilinear quark operator in field theory.

We consider a generalized version of the action for the charged scalar, including a dilaton-dependent mass term and a $\psi^4$ interaction term with a dilaton-dependent potential:

$$S_\psi = -M_p^3 N_c N_f \int d^5x \sqrt{-\det g}\, Z(\phi) \left[ |D_M \psi|^2 + M(\phi)^2 |\psi|^2 + \lambda_\psi(\phi) |\psi|^4 \right] , \qquad (22)$$

where the scalar field $\psi$ is vectorially charged

$$D_M \psi = \nabla_M \psi - i\hat{V}_M \psi . \qquad (23)$$

Note that we have fixed the charge of the scalar to one. This is because, similarly to the derivation of the Abelian DBI action (7) from (3), we interpret that $\psi$ arises as a Abelian component (i.e., the term proportional to unit matrix) of a more general flavored field $\Psi^{ij}$. This field transforms under the full non-Abelian chiral symmetry as $\Psi \mapsto V_L \Psi V_R$ where $V_{L/R} \in U(N_f)_{L/R}$, fixing the Abelian charge to one. Embedding the charged scalar in such a flavor matrix also means that its condensation breaks the full $U(N_f)$ vectorial chiral symmetry rather than just the baryon number. This breaking pattern is not similar to typical patterns expected in the paired phases of QCD, such as the color-flavor-locked phase, which breaks the axial $SU(N_f)$ instead. Nevertheless, we believe that this simple extension gives a reasonable estimate for the charge of the condensing field. Note also that the $\psi^4$ term in the action was not included in the original holographic superconductor models [94, 95]. This term is irrelevant for the phase diagram and the fluctuation analysis we carry out in Sec. 3, but it regularizes the probe limit computation of the condensate we carry out in Sec. 4. If we took the backreaction into account, $\psi$ would remain bounded even in the absence of this term so regularization would not be needed. We remark that the addition of the quartic term in the action is not motivated by simple considerations in field theory—since $\psi$ is interpreted to be dual to a quark bilinear, near the boundary this term is mapped to an operator with eight quark fields—but adding it is justified by generality as it is allowed by symmetry.

By varying of the action (22), the equations of motion for the charged scalar field is obtained

$$0 = D^M (Z(\phi) D_M \psi) - Z(\phi)(M(\phi)^2 + 2\lambda_\psi(\phi)|\psi|^2)\psi , \qquad (24)$$

where the equivalent equation for $\psi^*$ is given by the complex conjugate of (24). Inserting here the charged black hole background from (8) and (9) we obtain

$$\begin{aligned}
0 &= \psi'' + \left( 3A' + \frac{f'}{f} + (\log Z(\phi))' \right) \psi' + \eta^{\mu\nu} \partial_\mu \partial_\nu \psi + \\
&\quad + \frac{\Phi^2 - e^{2A} f M(\phi)^2}{f^2} \psi - \frac{2e^{2A}\lambda_\psi(\phi)}{f} |\psi|^2 \psi ,
\end{aligned} \qquad (25)$$

where primes denote derivatives with respect to $r$, and $\eta_{\mu\nu}$ is the Minkowski metric with mostly plus sign convention.

For the choices of the potentials in the $S_\psi$-sector, similar to the other sectors above, we also require the potentials to remain flat in the string frame at large $\phi$, by imposing

$$Z(\phi) \sim e^{2\phi} , \qquad M(\phi)^2 \sim e^{4\phi/3} \sim \lambda_\psi(\phi) . \qquad (26)$$

Therefore a simple Ansatz we can use is

$$
Z(\phi) = Z_0 \left( 1 + c_Z e^{2\phi} \right), \quad M(\phi)^2 = -C_0 \left( 1 + c_M e^{4\phi/3} \right), \quad \lambda_\psi(\phi) = L_0 \left( 1 + c_L e^{4\phi/3} \right),
\tag{27}
$$

where the constant parameters $Z_0$, $C_0$, $L_0$, $c_Z$, $c_M$, and $c_L$ specify the properties and extent of the paired phase in the phase diagram. Our conventions are such that the characteristic scale of the coupling $e^\phi$ in the RG flow (e.g. where one exits the perturbative near boundary region of the flow) is $e^\phi \sim 8\pi^2 \sim 100$. Consequently, natural choices for the values of the numerical coefficients $c_Z$, $c_M$, and $c_L$ are rather small.

According to the holographic dictionary, $\psi$ is dual to a charged scalar operator $\mathcal{O}_\psi$ whose vacuum expectation value serves as the order parameter. We follow a simplified approach where we interpret this operator to be a quark bilinear, so that its dimension equals three in the UV limit close to the boundary, which means setting $C_0\ell^2 = 3$ in the potentials. This is a simplification because expected quark bilinear order parameters for various paired phase in QCD are not gauge singlets. There are also gauge-invariant operators formed out of a higher number of quark fields whose expectation values work as order parameters [18], but since our aim here is not to construct a precise model for a specific superconductor phase, we find it more natural to interpret $\mathcal{O}_\psi$ as a diquark operator.

A trivial bulk profile for $\psi$ implies $\langle \mathcal{O}_\psi \rangle = 0$, corresponding to the normal quark matter phase. When $\psi$ develops a nontrivial profile, that is when it condenses, $\langle \mathcal{O}_\psi \rangle$ becomes nonzero, signaling a transition to the paired quark matter phase. We first analyze this condensation through identifying the corresponding instability in a fluctuation analysis in Sec. 3 (where we also discuss an instability towards a modulated phase), and then construct the end-point of the instability, i.e., the paired condensate, in the probe approximation in Sec. 4.

## 3  Stability

We discuss here two different types of instabilities of the chirally symmetric charged black hole background: pairing instabilities, which lead to the condensed quark matter phase, and spatially modulated instabilities, which lead to the inhomogeneous quark matter phase. Although these instabilities differ significantly in nature, they are both triggered by similar mechanisms stemming from the violation of the BF bound for the relevant fields at the zero temperature $AdS_2$ fixed point. To set the stage, we begin with a brief overview of the BF bound and the instability mechanisms it governs (for a more comprehensive review, see [56]). We remind that we are here studying the instabilities of the unpaired V-QCD background with $\psi = 0$, the condensed phase will be discussed in Sec. 4.

As already discussed in Section 2.2, the IR geometry of V-QCD is temperature-dependent. At $T = 0$, the geometry asymptotically approaches $AdS_2 \times \mathbb{R}^3$. As the temperature increases, the pure $AdS_2$ geometry is replaced by an $AdS_2$ black hole, and for $T \sim \Lambda$, the background becomes a "regular" charged black hole. This temperature-dependent structure naturally incorporates a mechanism for instabilities that can drive the system toward ordered phases.

In general, the most robust signal that a field is becoming unstable is the emergence of a complex IR scaling dimension at the $AdS_2$ fixed point, which is reached by the flows at zero temperature. This introduces a criterion on the bulk mass squared of the field, effectively setting a lower stability bound. This criterion is known as the BF bound [123–125]. According to the BF bound, a field with negative mass squared can still be stable, but it

becomes unstable once it drops below the critical threshold. Such instabilities are typically resolved by the condensation of the unstable field, and the resulting backreaction modifies the geometry, removing the instability in a self-consistent manner and leading to the emergence of an ordered phase at zero temperature.

As the system is heated up from zero towards higher temperatures, this picture also explains how the normal phase is restored above a critical temperature. As the temperature rises, the horizon moves inward along the holographic radial direction, effectively cutting off the IR region of the geometry. Eventually, the influence of the IR is diminished to the point where the negative mass squared of the field in question is insufficient to generate an instability, and the normal phase becomes stable.

While the mechanism described above is general, the nature of the ordering instability and its endpoint depend on the field in question. For the paired quark matter phase, the instability involves a charged scalar field, and the resulting condensation of a charged operator signals spontaneous breaking of the $U(1)$ baryon number symmetry. In contrast, the spatially modulated phase involves turning on a (non-)Abelian gauge field, leading to condensation of a (non-)Abelian chiral current operator and spontaneous breaking of translational symmetry. We analyze the AdS$_2$ BF bound explicitly in Appendix A, and show that it is violated both for charged scalar fluctuations (indicating an instability towards a paired phase) and modulated non-Abelian gauge field fluctuations (indicating an instability towards a modulated phase).

### 3.1 Pairing instability

We then analyze the range of the pairing instability by studying the quasinormal modes of the charged scalar fluctuations at finite frequency $\omega$ and momentum $k$. Plugging the plane wave Ansatz $\psi(t, \mathbf{x}, r) = e^{-i\omega t + i\mathbf{k}\cdot\mathbf{x}}\delta\psi(r)$ into the linearized form of (25), the corresponding fluctuation equation reads

$$
\begin{aligned}
0 &= \delta\psi''(r) + \left(3A'(r) + \frac{f'(r)}{f(r)} + \phi'(r)\frac{d}{d\phi(r)}\log Z(\phi(r))\right)\delta\psi'(r) + \\
&+ \frac{\Phi(r)^2 + \omega^2 - e^{2A(r)}f(r)M(\phi(r))^2 - k^2}{f(r)^2}\delta\psi(r)\,,
\end{aligned}
\tag{28}
$$

where $k = |\mathbf{k}|$. The quasinormal modes are found as the UV-normalizable solutions to this equation with infalling boundary conditions at the horizon, and can be computed numerically by following standard procedures (see, e.g., [126]).

At zero temperature, the emergence of a complex IR scaling dimension due to the violation of the BF bound for a charged scalar field implies that there is an infinite number of quasinormal modes in the upper half frequency plane, accumulating at $\omega = 0$ [56,99,127]. These modes are disallowed by causality and grow exponentially in time, thereby the system becomes unstable.

At nonzero temperature, only a finite number of unstable modes remain, and increasing the temperature finally stabilizes the system at a critical temperature $T_{\text{crit}}$. It can therefore be determined (as a function of the baryon chemical potential) by tracking the quasinormal modes with positive Im($\omega$). Actually, the relevant modes turn out to be purely imaginary, the instability is strongest at $k = 0$, and the modes are stabilized by crossing to the lower half-plane though the point $\omega = 0$.

At the critical temperature, $T = T_{\text{crit}}$, the last unstable mode becomes a zero mode. Therefore, $T_{\text{crit}}$ is determined by finding the normalizable solution to the $\omega = k = 0$ limit

of (28). With the choice of potentials given in (27), it is expressed as

$$
\begin{aligned}
0 \;=\; & \delta\psi''(r) + \left(3A'(r) + \frac{f'(r)}{f(r)} + \frac{2c_Z e^{2\phi(r)}\phi'(r)}{1 + c_Z e^{2\phi(r)}}\right)\delta\psi'(r) + \\
& + \frac{\Phi(r)^2 + \left(C_0 e^{2A(r)}f(r)\left(1 + c_M e^{\frac{4\phi(r)}{3}}\right)\right)}{f(r)^2}\delta\psi(r)\,.
\end{aligned}
\tag{29}
$$

Actually, one can count the number of unstable modes by using the (potentially unnormalizable) IR regular solution to (29). It equals the number of nodes in this solution. At each temperature, where a quasinormal mode passes from the upper half-plane to the lower half-plane, a node in the solution disappears by moving to the boundary. Therefore, to be precise, the critical temperature is determined by the normalizable nodeless solution to (29). This behavior is similar to that of eigenfunctions of a one-dimensional Schrödinger equation with a confining potential.

## 3.2  Modulated instabilities

As discussed above, spatially modulated instabilities originate from the same underlying mechanism as the pairing instability, albeit with additional complexity. In this case, the instabilities are driven by fluctuations of gauge fields, which are induced by the CS term (4). The most significant difference from the paired quark matter phase lies in the fact that the IR scaling dimension in this context depends nontrivially on the spatial momentum. It becomes complex, thus violating the corresponding BF bound, only for nonzero spatial momentum values ($k \neq 0$). Consequently, the condensate, involving chiral currents, develops a spatial modulation. In other words, the condensation of the gauge fields spontaneously breaks translational symmetry.

In the context of quark matter, spatially modulated instabilities were first explored within a top-down framework, namely the Witten-Sakai-Sugimoto model [128, 129]. These studies found that the instability region resides at high values of $\mu_b/T$ in the phase diagram. Only recently, the modulated instabilities has been studied in the V-QCD framework [101]. Remarkably, V-QCD predicts the onset of spatially modulated instabilities at much lower $\mu_b/T$ values, potentially even in the region relevant for the QCD critical end point. A follow-up study [102] further demonstrates that this low-$\mu_b/T$ prediction is largely model-independent. That is, the instability persists across a wide class of bottom-up holographic models fitted with the lattice data. This makes the resultant spatially modulated phase a natural competitor against the paired quark matter phase.

Here we briefly outline the procedure for determining the boundary of the instability region in the phase diagram. As shown in [102], both Abelian and non-Abelian fields lead to instabilities of this type, appearing in almost the same regions in the phase diagram. We focus here on the simpler non-Abelian case. The Abelian case, while qualitatively similar, involves additional complications due to its coupling with other fluctuation modes [130]. Nevertheless, our comparison in Section 5.3 with the paired instability will take into account both types of modulated instabilities.

We turn on non-Abelian gauge field fluctuations

$$
\delta A_{L/R}^M(x_M) = e^{-i(\omega t - kz)}\delta A_{L/R}^{Ma}(r)t^a \;.
\tag{30}
$$

on top of the finite density V-QCD background defined by equations (8) and (9). Here the momentum is chosen to be aligned with the $z$-direction, and $t^a$ are the SU($N_f$) generators. Because of chiral symmetry, the non-Abelian field fluctuations decouple from all the rest. It is enough to study the transverse components $\delta A_M^{L/R}$ with $M = x, y$, which receives

contributions from the CS term (4). The linearized fluctuation equations become diagonal in the $L/R$ basis,

$$
(\delta A_L^{\pm\,a})'' + \left[ \frac{d}{dr} \log e^A f w(\phi)^2 V_f(\phi) R \right] (\delta A_L^{\pm\,a})' +
$$
$$
+ \left[ \frac{\omega^2}{f^2} - \frac{k^2}{fR^2} \right] \delta A_L^{\pm\,a} \pm \frac{e^{-2A}\hat{n}k}{2\pi^2 f M_p^3 R^2 w(\phi)^4 V_f(\phi)^2} \delta A_L^{\pm\,a} = 0 \tag{31}
$$

where

$$
R = \sqrt{1 + \frac{\hat{n}^2}{e^{6A}w(\phi)^2 V_f(\phi)^2}} \; , \tag{32}
$$

and $\delta A_L^{\pm} = \delta A_L^x \pm i \delta A_L^y$. The right-handed fluctuations satisfy the same equation but with the opposite sign in the CS contribution.

As discussed above, equation (31) features a $k$-dependent effective mass term, which violates the BF bound only for $k \neq 0$ (see Appendix A.2 for details). This signals the presence of an instability with spatial modulation. The phase boundary is then determined by identifying the $\{T, \mu_q\}$ values at which a quasi-normal mode crosses into the upper half of the complex frequency plane.

## 4   Free energy and the condensate in the probe limit

Let us then analyze the end-point of the paired instability, i.e., the solution for the condensed phase in the probe approximation where the backreaction of the charged scalar in (22) to the geometry is neglected.

Assuming homogeneity and isotropy of $\psi = \psi(r)$ and inserting the potentials (27) the equation of motion (25) becomes

$$
\begin{aligned}
0 \;=\; & \psi'' + \left( 3A' + \frac{2c_Z e^{2\phi}\phi'}{1 + c_Z e^{2\phi}} + \frac{f'}{f} \right)\psi' - \frac{2L_0 e^{2A}\left(1 + c_L e^{\frac{4\phi}{3}}\right)}{f}\psi^3 + \\
& + \frac{\left(C_0 e^{2A} f \left(1 + c_M e^{\frac{4\phi}{3}}\right) + \Phi^2\right)}{f^2}\psi \,,
\end{aligned} \tag{33}
$$

where the coefficient $Z_0$ drops out and the fields $A, f, \phi$ and $\Phi$ are assumed to be known from a previously obtained background solution. We need to find the solution to this equation of motion which is regular at the horizon and normalizable at the UV boundary, since we are not turning on any source corresponding to the field $\psi$. Near the horizon $(r = r_h)$, the regular solution to (33) can be expressed in terms of a Taylor series

$$
\begin{aligned}
\psi(r) \;=\; & \sum_{n=0}^{\infty} \psi_n (r - r_h)^n \\
=\; & \psi_0 + \frac{\psi_0 e^{2A_0}}{f_1}\left(2L_0\psi_0^2\left(1 + c_L e^{\frac{4\phi_0}{3}}\right) - C_0\left(1 + c_M e^{\frac{4\phi_0}{3}}\right)\right)(r - r_h) + \mathcal{O}(r - r_h)^2 \,,
\end{aligned} \tag{34}
$$

where $A_0, f_1$ and $\phi_0$ are the coefficients of the analog series expansions for the background. Zero horizon values of $f$ and $\Phi$ imply $f_0 = \Phi_0 = 0$, where contributions of $\Phi$ only enter at orders larger than $\mathcal{O}(r - r_h)$. Provided initial conditions for $\psi_0$ the equation of motion (33) can then be solved by numeric integration from the horizon towards the boundary $r = 0$ and by imposing the desired asymptotic fall-off $\lim_{r\to 0}\psi(r)e^{A(r)} = 0$. In practice we

perform the numeric integration and determine $\psi_0$ using Mathematica's built in functions NDSolve and FindRoot, respectively. If there are several solutions, we pick the one with highest $|\psi_0|$. This solution has the lowest free energy.

In order to compute the free energy as function of the baryon chemical potential and temperature we numerically solve the action integral (1) for a given solution of the background and the scalar field $\psi$. Since backreaction is neglected, it is sufficient to determine the additive contribution of the condensate to the previously obtained contribution of the unpaired quark matter sector. The free-energy density $f_\psi$ is related to the pressure $p_\psi$ by

$$f_\psi(T, \mu) = T\, S_\psi^{\text{on-shell}} = -p_\psi(T, \mu)\,, \tag{35}$$

where $S_\psi^{\text{on-shell}}$ is the on-shell value of the action (22) (or to be precise, the corresponding density, after dividing out the volume of the space-time). Evaluating it for a solution of (24) with background fields of given chemical potential $\mu$ and temperature $T$ we obtain

$$S_\psi^{\text{on-shell}} = -M_p^3 N_c N_f \int dr Z_0 e^{5A(r)} \left(1 + c_Z e^{2\phi(r)}\right) \left[ -e^{-2A(r)} f(r)\psi'(r)^2 + \right.$$

$$\left. + C_0 \left(1 + c_M e^{\frac{4\phi(r)}{3}}\right)\psi(r)^2 + \frac{e^{-2A(r)}\Phi(r)^2\psi(r)^2}{f(r)} - L_0 \left(1 + c_L e^{\frac{4\phi(r)}{3}}\right)\psi(r)^4 \right] \tag{36}$$

$$= -M_p^3 N_c N_f \int dr Z_0 L_0 e^{5A(r)} \left(1 + c_Z e^{2\phi(r)}\right) \left(1 + c_L e^{\frac{4\phi(r)}{3}}\right)\psi(r)^4\,, \tag{37}$$

where we used the equation of motion (33) to obtain the latter expression. Unlike the equation of motion (24), the action integral explicitly depends on the parameter $Z_0$, thus introducing an additional degree of parameter dependence. We choose $Z_0$ such that the total entropy density vanishes in the limit of zero temperature $\lim_{T\to 0} \frac{\partial f}{\partial T}\Big|_\mu = 0$, where the total free energy density $f$ is the sum of the probe free-energy $f_\psi$ and the background free-energy.

## 5 Results

### 5.1 Phase diagram

The main result of our analysis is the phase diagram in Fig. 1. It shows the first order transition line separating the hadronic phase (including mesons and nuclear matter) and the deconfined quark matter phase constructed in [90]. The dashed second-order[3] transition line between paired and un-paired quark matter phases corresponds to the parameter values

$$c_M = 3.63 \cdot 10^{-2} \quad \text{and} \quad c_Z = 0\,, \tag{38}$$

while $Z_0$ is set to unity without loss of generality, because it drops out in this analysis. This parameter choice approximately maximizes the critical temperature of the condensed phase to $T_{\text{crit}} \approx 30\,\text{MeV}$ and at the same time ensures that it is never preferred in the vicinity of the critical point as expected from lattice QCD results for the cross over region at vanishing chemical potential. The critical temperature shows only mild dependence on the chemical potential. While there $T_{\text{crit}}$ decreases rapidly by about 20% from $\approx 40$ MeV to $\approx 30$ MeV with the baryon chemical potential right above the deconfinement transition, it remains almost constant and increases only slowly at larger values of $\mu_b$.

---

[3]In order to verify that the transition is of second order, in principle one should carry out the full backreacted analysis of the condensed phase. However, both earlier studies [94, 95] and the probe limit approach of Sec. 4 strongly suggest that the transition is quite in general of second order.

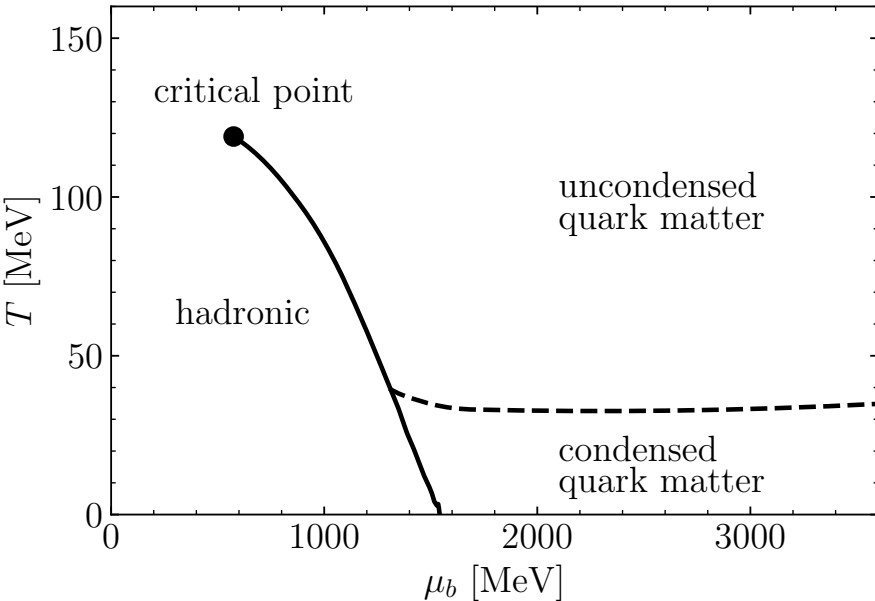

Figure 1: Hybrid V-QCD phase diagram in the baryon chemical potential and temperature plane. The solid line corresponds to the first-order deconfinement phase transition between the baryonic and quark matter sector where the dot indicates the critical end-point and the dashed line marks the second order transition between condensed and uncondensed quark matter.

Having presented our main result, we now explain how the parameter choice (38) was made. The procedure for determining the phase diagram in the presence of the pairing instability is described in Section 3.1. Here, we summarize how the phase boundary of the paired quark matter phase depends on model parameters. First, due to the linearization of the scalar field equation, the dependence on $\lambda_\psi$ drops out. Second, as shown in equation (28), the dependence on $Z_0$ also vanishes due to the structure of the equation of motion (24). We then choose $C_0$ such that the UV scaling dimension of the operator dual to $\psi$ field is three, i.e. $C_0\ell^2 = 3$. With this choice, the phase boundary of the pairing instability depends only on the parameters $c_M$ and $c_z$.

In general, the critical temperature $T_{\text{crit}}$ shows strong sensitivity to the values of $c_M$ and $c_z$. This is illustrated in Fig. 2, where three different parameter sets result in significantly different onset temperatures. Both $c_M$ and $c_z$ enhance the instability, i.e., increasing either of these parameters moves critical temperature upwards uniformly at all values of the chemical potential. However, their effects are slightly distinct: the influence of $c_M$ is stronger in the high-$\mu_b/T$ region.

Our objective is to determine the maximum value of $T_{\text{crit}}$, and thereby the maximal extension of the paired quark matter phase, while ensuring that it does not become favored near the critical point. Based on the previously summarized features of $c_M$ and $c_Z$, this objective is achieved by tuning these parameters in opposite directions: increasing $c_M$ enhances the instability at high $\mu_b/T$ while decreasing $c_Z$ to compensates the associated enhancement in low-$\mu_q/T$ region thus maintaining consistency with the lattice data. This tuning strategy can be seen in Fig. 2, where larger values of $c_M$ raise the critical temperature $T_{\text{crit}}$ (shown as bold dashed curves), while decreasing values of $c_Z$ keeps the low-$\mu_b/T$ region unaffected (shown as transparent dashed curves). Once this tuning strategy is implemented and optimized using an automated routine, the phase boundary converges its maximal extent in the high-$\mu_b/T$ region. This leads to the parameter values

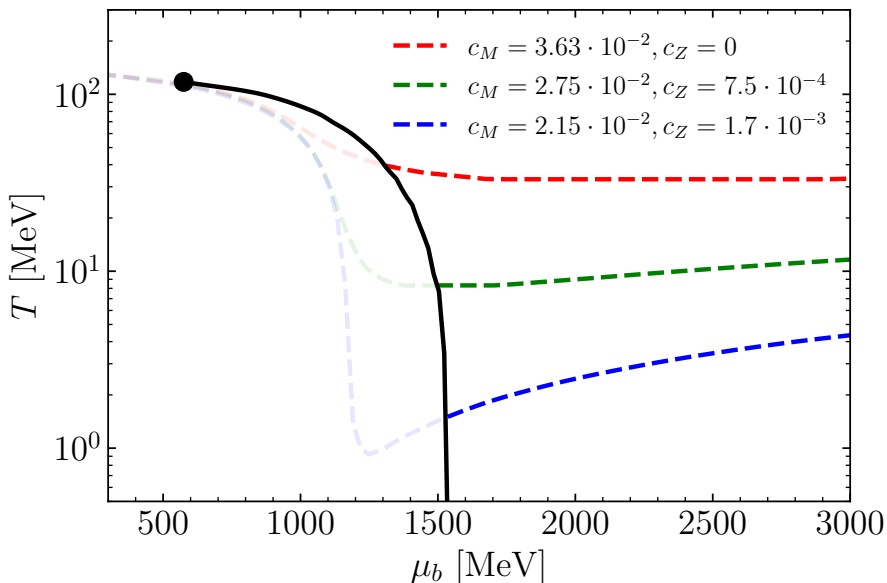

Figure 2: Parameter dependence of the condensed QM phase on the temperature and baryon chemical potential plane. Also shown in light colors is the extent of the condensate on the left of the first-order deconfinement transition (black line), where the deconfined phase is replaced by the thermodynamically favored confined baryon phase in the hybrid model.

$c_M = 3.63 \cdot 10^{-2}$ and $c_Z = 0$, which produce the phase diagram shown in Fig. 1, the main result of this work.

Finally, we emphasize that the phase boundary is insensitive to the backreaction the scalar field on the background geometry and other sectors. This is justified by the fact that the phase transition is of second order. Therefore the scalar field remains perturbatively small near the transition, rendering its backreaction negligible.

## 5.2  Equation of state

Here, we present the leading-order contribution of the paired phase to the equation of state, specifically the pressure as a function of temperature and chemical potential (see Sec. 4). Unlike the phase transition lines discussed in the previous section, a rigorous computation of the equation of state would require accounting for the backreaction of the scalar field in the paired phase on the other sectors. In this sense, the results in this section should be viewed as only a leading-order correction to the fully backreacted result for the pressure as a function of temperature and chemical potential in the paired phase.

In Fig. 3 we report our results for the pressure, with and without quark pairing, as a function of temperature at three different fixed values for the baryon chemical potential. The results correspond to the phase diagram in Fig. 1, i.e., to the values for $c_M = 3.63 \cdot 10^{-2}$ and $c_Z = 0$ that maximize the value of the critical temperature and set $c_L = (8\pi^2)^{-4/3}$ while assuming $L_0 = 1$. Here the choice of $c_L$ reflect the characteristic scale of the coupling, $e^\phi \sim 8\pi^2$. Additionally, we fix $Z_0 \approx 2.5$ to ensure that the total entropy $S = \frac{\partial P}{\partial T}$, given as the sum of the probe contribution from the scalar and the background entropy, vanishes at zero temperature. In the leading-order approximation, this adjustment is necessary to ensure thermodynamic stability by compensating for the missing backreaction, which becomes relevant in the low-temperature limit. With these adjustments, we arrive at an approximate 10% increase for the pressure at temperatures $T < 5$ MeV. As expected,

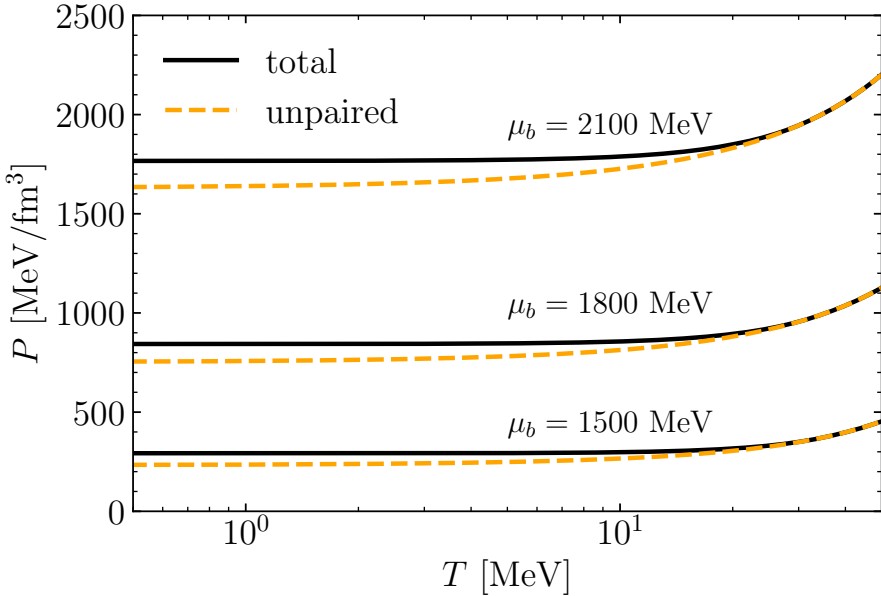

Figure 3: Total pressure (black) and unpaired QM pressure (dashed orange) for fixed $\mu_b = 1500, 1800, 2100$ MeV (bottom to top) and the the parameter set $c_M = 3.63 \cdot 10^{-2}$ and $c_Z = 0$ corresponding to the phase diagram of Fig. 1, while setting $c_L = (8\pi^2)^{-4/3}$ and $L_0 = 1$.

the pressure of the paired phase converges to the one of the unpaired phase at the critical temperature.

## 5.3   Stability analysis

Finally, we report detailed results of our stability analysis. Figure 4 is a summary of the stability lines determined by $\text{Im}(\omega)$ for the unstable modes as a function of temperature for various fixed values of the chemical potential, which characterizes how fast the instability grows at early times after it is seeded. Each panel displays three stability lines,

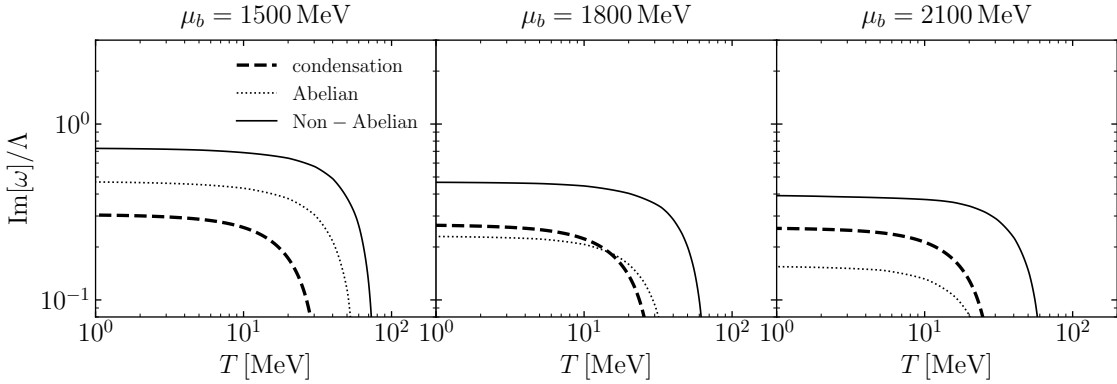

Figure 4: The imaginary part of the frequency of the most unstable mode as a function of temperature is shown for three fixed values of the chemical potential (left to right) for the non-Abelian, Abelian, and pairing instabilities, represented by solid, dotted, and dashed lines, respectively.

corresponding to the thresholds for the formation of the spatially modulated phases in

the Abelian and non-Abelian sectors of the model discussed in Sec. 3.2, as well as for the condensate of the paired phase introduced in Sec. 3.1. The amplitude of each curve represents the strength of the instability, providing a measure to identify the strongest instability leading to the formation of the corresponding phase.

To construct the stability lines for the spatially modulated phases, we first fixed the value of the chemical potential. Then for each temperature value, we constructed the V-QCD background and solved the fluctuation equations (either the non-Abelian equation (31) or the Abelian equations given in [101]) to look for modes that show an instability, which means that they have a positive imaginary part. Then we proceed to find at which value of the momentum the imaginary part of the frequency of the unstable mode reached a maximum value. This maximum value of $\mathrm{Im}(\omega)$, for a given temperature is then a measure of the strength of the instability. This is because a higher value of $\mathrm{Im}(\omega)$ means that the plane wave eigenmode of the QNM will grow faster in the temporal evolution of the system.

Overall, we find that the non-Abelian instability is the most dominant across the range of chemical potentials and temperatures considered. Although the modulated instabilities become weaker with increasing chemical potential, the non-Abelian instability always dominates over the instability towards paired quark matter condensation. This is also consistent with the analysis of the BF bound violation in Appendix A: the violation was most severe for the non-Abelian instability. Interestingly, around $\mu_b \approx 1.8$ GeV, the Abelian and paired instabilities become comparable in strength, and at larger values of $\mu_b$, the latter dominates over the former. In summary, our stability analysis suggests that the homogeneous condensed phase, constructed here in the probe approximation, does not represent the true ground state of the model. Instead, the preferred state is inhomogeneous non-Abelian phase or a mixture of the inhomogeneous (gauge field) and paired (charged scalar) condensates.

# 6   Conclusion

Our work extends the V-QCD model by introducing a sector that facilitates the condensation of a charged scalar field in the quark phase, mimicking the pairing of quark matter expected in color-superconducting states in QCD. We present the phase diagram of the extended model, which includes the previously computed first-order deconfinement transition along with the second-order transition line for the paired phase, as determined in this work. By tuning the two relevant parameters of the model, we arrive at an estimated upper bound for the critical temperature for the condensation of paired quark matter of about $T_{\mathrm{crit}} \approx 30$ MeV, which depends only mildly on the chemical potential in the range $\mu_b = 1.5 - 3$ GeV. Furthermore, we find that the leading-order contribution to the pressure in the paired phase is approximately 10% higher than in the unpaired phase. Finally, our stability analysis, which compares the stability lines of paired quark matter and modulated quark matter, suggests that the modulated phase is preferred.

Clearly, the analysis presented in this work represents only an initial step toward a phenomenologically relevant modeling of quark pairing and, ultimately, color superconductivity in V-QCD with potential applications to neutron star physics. Since paired phases increase the pressure in the deconfined phase compared to when neglecting pairing effects, we expect this to enhance the support against gravitational collapse in isolated stars and possibly lead to stable stars with paired quark matter cores as found in [84]. This would lead to higher maximal masses and a prolonged lifetime of the post-merger remnant, potentially accompanied by a modified amount of quark matter produced during

the merger. Such changes are likely to impact, for example, the upper limit of neutron star masses, possibly altering the quasi-universal ratio between maximum masses of rotating and non-rotating stars [87, 131], the compatibility of softer variants of the V-QCD EOS family with the observed lifetime of the GW170817 remnant [92], and the critical mass threshold for prompt collapse [37].

Several important improvements are necessary to further develop this approach to determine the aforementioned implications on neutron stars. A natural first step is to incorporate the backreaction of the paired phase. On the gravity side, the backreacted charged scalar acts as a source for the black hole charge, removing the flux that supports the $AdS_2$ geometry, therefore automatically leading to zero entropy. This would not only eliminate the need for the ad hoc tuning of the model parameter $Z_0$ to ensure vanishing entropy at zero temperature in the paired phase but also enable a rigorous computation of the cold pressure in this phase. Consequently, this would lead to corrections in the deconfinement transition line. This improvement is particularly significant because the previously used unpaired quark matter version of the model does not support stable quark matter in isolated stars, but only in binary neutron star merger remnants [92, 93]. We should also check that the transition between paired and unpaired quark matter is of the second order also after taking the backreaction into account. The second issue is to extend the superfluid-like description constructed in this work for V-QCD to include color Higgsing, that is, to generalize the spontaneous breaking of the global $U(1)$ baryon number symmetry to the breaking of local $SU(N_c)$ color group, in a manner similar to what was done in [74]. A third issue that needs to be addressed is the neglect of quark masses in the model. As demonstrated in [102], extending the model to include three quark flavors can significantly alter its phase structure. Future work will focus on investigating how such more realistic extensions influence quark matter condensation. Finally, we wish to establish an updated overall state-of-the-art holographic model of QCD at finite temperature and density, which includes all these new developments, significantly improves the existing EOS models [90], and can also be used to derive predictions for transport. Among other things, such an overall model can be used as an input in neutron star merger simulations in order to analyze the aforementioned effects of quark pairing.

# Acknowledgements

We are thank Jürgen Schaffner-Bielich, Michael Buballa, Hosein Gholami, Tyler Gorda, Umut Gürsoy, Marco Hofmann, Niko Jokela, Nicolas Kovensky, Edwan Préau, Ishfaq Ahmad Rather, Luciano Rezzolla, Matthew Roberts, and Andreas Schmitt for useful discussions and comments on our manuscript.

**Funding information**   JCR was supported by a DGAPA-UNAM postdoctoral fellowship. TD has been supported by the Netherlands Organisation for Scientific Research (NWO) with the grants VICI–BN.000665.1 and ENW XL–BN.000704.1. CE acknowledges support by the DFG through the CRC-TR 211 "Strong-interaction matter under extreme conditions" – project number 315477589 – TRR 211. MJ has been supported by an appointment to the JRG Program at the APCTP through the Science and Technology Promotion Fund and Lottery Fund of the Korean Government and by the Korean Local Governments - Gyeongsangbuk-do Province and Pohang City - and by the National Research Foundation of Korea (NRF) funded by the Korean government (MSIT) (Grant no. 2021R1A2C1010834).

# A    Instabilities at the zero temperature $AdS_2$ point

In the limit of zero temperature, if the chemical potential is high enough, the black hole solutions in V-QCD become extremal and the IR geometry approaches the form $AdS_2 \times \mathbb{R}^3$ up to logarithmic corrections [77, 110]. The fixed point $AdS_2 \times \mathbb{R}^3$ geometry is also an exact solution of the V-QCD action, found by inserting an Ansatz with constant values of $A_*$ and $\phi_*$. The equation of motions imply

$$V_{\text{eff}}(\phi_*, A_*, \hat{n}) = 0 = \frac{\partial V_{\text{eff}}(\phi_*, A_*, \hat{n})}{\partial \phi} \ , \tag{39}$$

where the effective potential is defined by

$$V_{\text{eff}}(\phi, A, \hat{n}) \equiv V_g(\phi) - x V_f(\phi)\sqrt{1 + \frac{\hat{n}^2}{e^{6A} V_f(\phi)^2 w(\phi)^2}} \ . \tag{40}$$

These equations determine the value of $\phi_*$ and the combination

$$\tilde{n}_* = \frac{\hat{n}}{e^{3A_*}} \ , \tag{41}$$

while $A_*$ remains a free parameter, linked to the (logarithm of the) IR energy scale. The resulting geometry is

$$ds^2 = \frac{L_2^2 dr^2}{r^2} - \frac{e^{4A_*} r^2 dt^2}{L_2^2} + e^{2A_*} d\mathbf{x}^2 \tag{42}$$

where the $AdS_2$ radius is

$$L_2 = \frac{1}{\sqrt{\frac{1}{6} \frac{\partial V_{\text{eff}}(\phi_*, A_*, \hat{n})}{\partial A}}} \ . \tag{43}$$

## A.1    BF bound for the pairing instability

We then determine the BF bound for fluctuations of $\psi$ around the $AdS_2$ geometry for V-QCD. To this end, one just needs to consider the fluctuation equation (28) in the limit of zero frequency and momentum, and insert the $AdS_2$ solution in (42). This gives

$$\delta\psi''(r) + \frac{2\delta\psi'(r)}{r} + \frac{e^{-4A_*} L_2^2 \left(-e^{4A_*} M(\phi_*)^2 r^2 + L_2^2 \Phi^2\right)}{r^4} \delta\psi(r) = 0 \ . \tag{44}$$

Because $\Phi$ vanishes at the horizon, its equation of motion implies that $\Phi \approx C_1 r$ as $r \to 0$. The coefficient $C_1$ can be determined from the equation of motion to be

$$C_1 = \frac{n_*}{w(\phi_*)^2 V_f(\phi_*)\sqrt{1 + \frac{n_*^2}{w(\phi_*)^2 V_f(\phi_*)^2}}} \ . \tag{45}$$

Then inserting a power law Ansatz $\psi = r^{-\Delta_*}$ to determine IR scaling dimension, the fluctuation equation gives

$$\Delta_* = \frac{1}{2}\left(1 \pm \sqrt{1 - 4C_1^2 L_2^4 + 4L_2^2 M(\phi_*)^2}\right) \ . \tag{46}$$

The BF bound is found by requiring that the scaling dimension is real. Note that the mass squared $M(\phi)^2$ is the only parameter from the action of $\psi$ appearing here, as the

other coefficients are determined by the background. In terms of the mass, the BF bound becomes

$$M(\phi_*)^2 \geq C_1^2 L_2^2 - \frac{1}{4L_2^2} \approx -2.42817 \ , \tag{47}$$

where the numerical value is for the potentials 7a specified in Sec. 2.

If we choose $M(\phi)^2$ such that the BF bound is violated, the solution for $\psi$ oscillates as one approaches the $AdS_2$ endpoint. That is, in the zero temperature limit the solution has an infinite number of nodes, indicating the presence of an instability.

The value of $M(\phi_*)^2$ with the parameter values given in (38) is $M(\phi_*)^2 = -3.14596$ and the corresponding IR scaling dimension is $\Delta_* = 0.5 \pm 0.2617i$.

## A.2  BF bound for the non-Abelian modulated instability

Let us then analyze the BF bound for the case of the non-Abelian inhomogeneous instability, governed by the fluctuation equation in (31). This equation can be rewritten as

$$(\delta A_L^{\pm\,a})'' + \left[\frac{d}{dr} \log e^A f w(\phi)^2 V_f(\phi) R\right] (\delta A_L^{\pm\,a})' + \frac{\omega^2}{f^2} \delta A_L^{\pm\,a} +$$
$$- \frac{1}{fR^2} \left[\left(k \mp \frac{e^{-2A}\hat{n}}{4\pi^2 M_p^3 w(\phi)^4 V_f(\phi)^2}\right)^2 - \left(\frac{e^{-2A}\hat{n}}{4\pi^2 M_p^3 w(\phi)^4 V_f(\phi)^2}\right)^2\right] \delta A_L^{\pm\,a} = 0 \ , \tag{48}$$

inserting here the $AdS_2$ solution and considering $\omega = 0$ limit

$$(\delta A_L^{\pm\,a})'' + \frac{2}{r}(\delta A_L^{\pm\,a})'$$
$$- \frac{L_2^2}{r^2 R_*^2} \left[\left(\frac{k}{e^{A_*}} \mp \frac{e^{-3A_*}\hat{n}_*}{4\pi^2 M_p^3 w(\phi_*)^4 V_f(\phi_*)^2}\right)^2 - \left(\frac{e^{-3A_*}\hat{n}_*}{4\pi^2 M_p^3 w(\phi_*)^4 V_f(\phi_*)^2}\right)^2\right] \delta A_L^{\pm\,a} = 0 \ . \tag{49}$$

One can then proceed by plugging a power law Ansatz $\delta A_L^{\pm\,a} = r^{-\Delta_*}$ into (49) to determine $k$-dependent IR scaling dimension. However, the value maximizing the effect already can be read off from (49)

$$\frac{k}{e^{A_*}} = \pm \frac{\tilde{n}_*}{4\pi^2 M_p^3 w(\phi_*)^4 V_f(\phi_*)^2} \approx \pm 6.9268 \ . \tag{50}$$

This maximizing value leads to the IR scaling dimension with a value of $\Delta_* \approx 0.5 \pm 0.3823i$.

## A.3  Rough bound for the critical temperature

Note that the BF-bound violating dimensions at the IR fixed point were of the form $\Delta_* = 1/2 \pm \nu i$ above. This means that the fluctuation wave function oscillates as

$$\delta\psi \propto \sin(\nu \log r) \tag{51}$$

as $r \to 0$. This allows us to estimate at which temperatures the instability is unavoidable. Namely the finite-temperature generalization of the geometry (42) is obtained by modifying the blackening factor to

$$f(r) \approx \frac{e^{2A_*}r^2}{L_2^2} \left(1 - \frac{r_h}{r}\right) \tag{52}$$

if $r_h$ is sufficiently small. The temperature is given by

$$T = \frac{1}{4\pi} f'(r_h) \approx \frac{1}{4\pi} \frac{e^{2A_*} r_h}{L_2^2} \; . \tag{53}$$

The geometry takes the AdS$_2$ form for $r_h \lesssim r \lesssim c_f e^{-A_*}$ where the upper limit indicates where the flow away from the fixed point starts to modify the geometry significantly. We expect that the coefficient $c_f$ is of the order of one so we can drop it in order to obtain a rough estimate. A node in (51) is unavoidable in the AdS$_2$ range if $-\nu \log(r_h e^{A_*}) > \pi$. This translates to

$$T_{\text{crit}} e^{-A_*} \gtrsim e^{-\frac{\pi}{\nu}} \; , \tag{54}$$

where we dropped $\mathcal{O}(1)$ factors. While the derivation gives a lower bound, we expect that the critical temperature shows a similar scaling as this bound. That is, if $\nu$ is small, the instability is restricted to exponentially small temperatures. In the context of our model, this means that a substantial violation of the bound is required for the instability to reach temperatures found in neutron stars and neutron star mergers.

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
