# Peer review of "Towards holographic color superconductivity in QCD"

_SciPost Physics_

## Round 1 · Referee Report · Anonymous (Referee 1) · 2025-8-20

Strengths

1- Novel introduction of framework to mimic quark pairing in the V-QCD model. 2- New phase diagram that includes the paired-phase transition line. 3- Leading order contribution of the pressure for the paired phase. 4- Supports the existence of high-mass quark stars that survive gravitational collapse due to quark pairing mechanism. 5- Outlines the pathway/steps to further improve the model, e.g. take into account the backreaction of all fields in the equations of motion.

Weaknesses

1- Although the authors claim their model agrees with lattice QCD thermodynamics, no figure/plot is provided in the article. 2- Full backreaction of fields need to be taken account to be sure the resulting equation of state describes lattice QCD data.

Report

The Journal's acceptance criteria are met. The authors describe the added framework for quark pairing in the deconfined phase diagram within the V-QCD model that already includes a first order transition line. The novel unpaired-to-paired quark matter transition line is conjectured to be a second order. I have some comments/questions regarding the manuscript that do not affect in any way the acceptance decision, but readers might benefit from.

1) In Section 3.2 the authors claim that spatially modulated instabilities, which appear at high $\mu_{B}/T$ in the Witten-Sakai-Sugimoto model and at lower $\mu_{B}/T$ in V-QCD, are model independent and persist across a wide class of bottom up holographic models fitted with lattice data. Do these ones also include the Einstein-Maxwell-dilaton models described, for example in Rougemont et al. "Hot QCD phase diagram from holographic Einstein–Maxwell–Dilaton models" Prog.Part.Nucl.Phys. 135 (2024) 104093? Are these instabilitities a feature of holographic models? Are they also expected to appear in QCD?

2) In Section 5.1, the authors describe the phase diagram from Fig. 1 that shows a first order transition line. Does the corresponding equation of state (EoS) agree with lattice QCD data from zero to small $\mu_{B}/T$? Reference [90] does not show a comparison between lattice QCD and the resulting EoS from the effective model either. Also, does the inclusion of the quark-pairing mechanism affect the agreement between V-QCD and lattice QCD?

Requested changes

1- Figures that showcase comparisons between the V-QCD EoS from this novel framework and lattice QCD data are highly recommended to be included in the manuscript, similarly to what has been done in Ref. [119]

Recommendation

Publish (easily meets expectations and criteria for this Journal; among top 50%)

  • validity: top
  • significance: top
  • originality: top
  • clarity: top
  • formatting: excellent
  • grammar: excellent

Author:  Christian Ecker  on 2025-10-14  [id 5926]

(in reply to Report 1 on 2025-08-20)
Category:
answer to question

Dear Editor and Anonymous Referee,

We would like to thank you for the professional handling of our manuscript and for recommending it for publication in your journal. Below, we address the referee’s comments and summarize the revisions made to our manuscript.

Best regards, Christian Ecker, on behalf of all authors

Referee: 1) In Section 3.2 the authors claim that spatially modulated instabilities, which appear at high μB/T in the Witten-Sakai-Sugimoto model and at lower μB/T in V-QCD, are model independent and persist across a wide class of bottom up holographic models fitted with lattice data. Do these ones also include the Einstein-Maxwell-dilaton models described, for example in Rougemont et al. "Hot QCD phase diagram from holographic Einstein–Maxwell–Dilaton models" Prog.Part.Nucl.Phys. 135 (2024) 104093? Are these instabilitities a feature of holographic models? Are they also expected to appear in QCD?

Reply: Indeed, the generic emergence of modulated instabilities in both V-QCD and Einstein–Maxwell–Dilaton (EMD) bottom-up models tuned to QCD has been demonstrated in [arXiv:2405.02399] and [arXiv:2405.02392]. Such instabilities are expected to be a common feature of holographic models, including those presented in “Hot QCD phase diagram from holographic Einstein–Maxwell–Dilaton models” (Prog. Part. Nucl. Phys. 135 (2024) 104093), which share key characteristics with the aforementioned EMD frameworks. That is, while the extent of the instability was not checked for precisely the fits discussed in the review by Rougemont et al., it was checked for similar fits by other authors in [arXiv:2405.02392], and the dependence on the details of the fit was observed to be so small that it is clear that the main findings also apply to the models of this review.

However, we stress that the class of models to which our comment applies is limited to (chirally symmetric) holographic models precisely fitted to lattice data for the equation of state and baryon number susceptibilities of QCD. In particular, while there are indications for the existence of inhomogeneous phases in QCD, there are several factors which make a direct quantitative connection of the holographic studies to real (N=3) QCD challenging. These include i) effects of chiral symmetry breaking through quark masses ii) effects missed due to the inherently large-N limit of holographic studies iii) similarly, effects missed due to the inherently infinite coupling.

Referee: 2) In Section 5.1, the authors describe the phase diagram from Fig. 1 that shows a first order transition line. Does the corresponding equation of state (EoS) agree with lattice QCD data from zero to small μB/T? Reference [90] does not show a comparison between lattice QCD and the resulting EoS from the effective model either. Also, does the inclusion of the quark-pairing mechanism affect the agreement between V-QCD and lattice QCD?

Reply: For simplicity, our manuscript focuses on the parameter set 7a introduced in [arXiv:1809.07770], which represents a canonical intermediate choice. This set not only lies well within the constrained band of model-agnostic parametrizations at zero temperature (see, e.g., Fig. 2 in [arXiv:2402.11013])—being sufficiently soft to satisfy the tidal deformability constraints inferred from the GW170817 event—but is also stiff enough to prevent a prompt collapse and to allow for an extended, potentially second-long-lived post-merger remnant, as suggested by the EM counterpart observed in this event. Indeed, the fit to lattice data corresponding to this specific parameter set is not shown in our manuscript but can be found in Fig. 1 (Yang–Mills sector only) and Fig. 2 (full model) of [arXiv:1809.07770]. In Section 2.3, we have added a new plot demonstrating the consistency of the thermodynamic properties of the model obtained from potential choice 7a with the 2+1 flavor lattice data from the Wuppertal–Budapest and HotQCD collaborations. Since we constrain the model using lattice QCD data at small baryon chemical potential and high temperature—where quark pairing is not realized—pairing is not expected to affect the parameter fit. Furthermore, we note that the present study does not include the backreaction of the condensate on the gluon and flavor sectors. Consequently, at this level of approximation, the paired phase cannot influence the fit to lattice data.

Requested changes: 1) Figures that showcase comparisons between the V-QCD EoS from this novel framework and lattice QCD data are highly recommended to be included in the manuscript, similarly to what has been done in Ref. [119]

Reply: We agree with the referee that including a direct comparison of the lattice data with the V-QCD model used will improve the manuscript and make it more self-contained. We have therefore added a new Figure 1 in Section 2.3, which now shows a comparison of the interaction measure, pressure, and baryon number susceptibility at zero chemical potential as functions of temperature, as obtained from the V-QCD 7a model fit used throughout the manuscript, together with the corresponding lattice data from the Wuppertal–Budapest and HotQCD collaborations.

---

## Round 1 · Referee Report · Anonymous (Referee 2) · 2025-10-31

Strengths

1) Improvement of the V-QCD model by including a scalar sector that is aimed to model quark matter pairing 2) Estimate of the upper bound for the critical temperature 3) Computation of the equation of state at leading order 4) Stability analysis comparing the spatially modulated phase with the paired quark phase

Weaknesses

1) The modelization with a scalar field gives a symmetry breaking pattern that is not the one expected in the paired phase of real QCD 2) Several results are dependent on choices (especially for the potentials) that are not fully justified 3) The absence of a backreacted analysis prevents to be conclusive about the order of the transition

Report

The authors extend the holographic V-QCD model by including a scalar sector that mimicks the condensation of paired quarks. They obtain a phase diagram with a phase transition between paired and unpaired (deconfined) quarks, and argue that this is second-order. The instability towards the paired phase is ultimately subdominant with respect to the instability towards a modulated phase.

The paper is nicely written and provides a futher step in the direction of studying quark pairing in finite density and finite temperature holographic QCD. However, I have some comments and remarks about the manuscript, that I would like the authors to address, especially points 4) and 8).

1) Around eq. (3) the authors assume they can ignore the tachyon field dual to the gauge-invariant quark bilinear, because the latter has a vanishing vev in the deconfined phase. Shouldn't this be a consequence of the analysis rather than an assumption? It is not clear if this is the case or not.

2) The combination on the LHS of eq. (11) seems to be obtained by summing eqs. (2) and (7), and neglecting the term proportional to F_MN in (7). However, it is not clear at this stage that this corresponds to the weakly-coupled limit where \phi goes to -\infty, as the explicit form of w(\phi) is given later in eq. (14). Is there a simple argument for this?

3) The authors make some choices about the potentials that would require some clarifications. These are: - a choice of a potential set in eqs. (15-17) among a one-parameter family that is allowed by constraints by lattice data - a choice of asymptotic behavior of Vf in eq. (16) that departs from the string-frame flat result of eq. (14) - a choice of a free parameter (namely, the AdS radius, or W0) - a choice of a potential set in (27) It is not clear why the authors made these choices, and if and how they affect (at least qualitatively) their final results. It would be worthy to comment about this in section 2.3.

4) After eq. (23) the authors discuss that if the scalar is embedded in a flavor bifundamental, then no non-zero vev would preserve the full U(N) vectorial symmetry. The added scalar is only charged under the vector U(1), so it seems that they are describing a deconfined phase where baryon number is spontaneously broken, rather than a paired phase. It is then not clear that the phase they discuss should be phenomenologically relevant for QCD. Moreover, as they correctly mention after eq. (27), the scalar cannot be a naive diquark operator, as this is not gauge invariant. It is then not clear why they do not interpret instead the scalar as a higher-order gauge-invariant quark operator (this would change the UV scaling dimension from 3 to a higher one, though).

5) Are there any lattice/phenomenological/holographic results confirming the fact that a spatially modulated phase is favored with respect to a paired quark phase, as the results of section 3 suggest?

6) Why does the probe analysis of section 4 suggest the transition to be second order? This is mentioned in footnote 3, but in principle it would require to verify the continuity of the (derivative of the) free energy at the transition. What is the argument of the authors in favor of this?

7) At the end of section 5.1 the authors claim that the backreaction of the scalar field is negligible, based on the fact that the transition is second order. However, they also claim that the transition should be second order, if the backreaction is neglected. Since the backreaction has not been computed, it is not clear what is an assumption and what has been verified.

8) As a consequence of the stability analysis, the authors observe that the modulated phase is favored with respect to the paired phase. It is not clear then how the phase diagram in Figure 1 should be interpreted, given that the modulated phase is favored with respect to the homogeneous paired phase. In particular, do the authors expect this condensed quark matter to exist in their model? How does this compare with real QCD expectations?

Requested changes

1) Clarify the choice of potentials and its consequences.

2) Clarify why the interpretation of the scalar as a diquark is valid.

3) Clarify what are the arguments in favor of the transition being second order.

4) Clarify the phase structure of the theory and the role of the modulated phase with respect to the one shown in figure 1.

Recommendation

Ask for minor revision

---

## Editorial Decision

in_refereeing